# Grounding the Body Improves Sleep Quality in Patients with Mild Alzheimer’s Disease: A Pilot Study

**DOI:** 10.3390/healthcare10030581

**Published:** 2022-03-20

**Authors:** Chien-Hung Lin, Shih-Ting Tseng, Yao-Chung Chuang, Chun-En Kuo, Nai-Ching Chen

**Affiliations:** 1Department of Chinese Medicine, Kaohsiung Chang Gung Memorial Hospital, College of Medicine, Chang Gung University, Kaohsiung 833, Taiwan; b9505027@cgmh.org.tw (C.-H.L.); b9705017@cgmh.org.tw (S.-T.T.); 2Department of Neurology, Kaohsiung Chang Gung Memorial Hospital, College of Medicine, Chang Gung University, Kaohsiung 833, Taiwan; ycchuang@ms87.url.com.tw; 3School of Chinese Medicine for Post Baccalaureate I-Shou University, No. 1, Sec. 1, Syuecheng Rd., Dashu District, Kaohsiung 84001, Taiwan; 4Department of Leisure and Sports Management, Cheng Shiu University, No. 840, Chengcing Rd., Niaosong Dist., Kaohsiung 83347, Taiwan

**Keywords:** dementia, grounding, Pittsburgh Sleep Quality Index, Beck Anxiety Inventory, Beck Depression Inventory, complementary and alternative therapy, insomnia

## Abstract

Background: Grounding refers to having direct contact with the Earth, such as by walking barefoot or lying on the ground. Research has found that grounding can improve inflammation, free radical damage, blood pressure, sleep quality, pain, stress, mood, and wound healing. However, there has been no research on the effect of utilizing grounding for patients with Alzheimer’s disease (AD). Thus, in this study, we investigated the effectiveness of grounding as a non-pharmacological therapy for treating sleep disturbances, anxiety, and depression in patients with mild AD. Methods: Patients with mild AD were enrolled in the study. The electrochemical analyzer CHI 1205b was employed to check the electrochemical signals at acupoints KI1 and GV16. We used the Pittsburgh Sleep Quality Index (PSQI), Beck Anxiety Inventory (BAI), and Beck Depression Inventory-II (BDI-II) to evaluate sleep quality, anxiety, and depression, respectively, at weeks 0 and 12. Results: This 12-week placebo-controlled study enrolled 22 patients, but only 15 patients completed the 12-week intervention and survey. Grounding significantly improved PSQI scores compared to the sham-grounding group (mean ± SD: 0.3 ± 0.7 vs. 3.0 ± 1.9, *p* = 0.006). The scores on the BAI and BDI-II did not change significantly after grounding in comparison to the sham-grounding group. Conclusions: Grounding can improve sleep quality, but it does not significantly improve anxiety and depression among patients with mild AD.

## 1. Introduction

Alzheimer’s disease (AD) is the most prevalent neurodegenerative disease that causes dementia by impairing mental growth capacity and disrupting neurocognitive activity [1]. Despite recent advancements in AD therapy, therapeutic effectiveness has been small, noncurative, and susceptible to drug resistance. Behavioral and psychological symptoms (BPSD) are the most troublesome problems to take care of during disease progression [2]. BPSD represents a heterogeneous group of sleep, affective, psychotic, and behavioral symptoms (sundowning, excessive daytime sleepiness, nocturnal wandering) [3] that occur in the majority of patients with dementia, causing great suffering and increasing the burden on caregivers [4]. In fact, poor sleep in persons with AD is the most common stressor for family caregivers [5]. To treat agitation, aggressiveness, delusions, and hallucinations in patients with AD, non-pharmacological approaches are usually preferred, and the use of antipsychotic drugs should be limited due to the increased risk of mortality, stroke, and hallucination, as well as a higher risk of relapse after discontinuation [4]. However, behavioral symptoms are not always controlled by non-pharmacological interventions. Therefore, atypical antipsychotics are commonly prescribed for the treatment of behavioral symptoms [1].

Grounding, or earthing, refers to contact of the body with the ground, such as by walking barefoot, which can affect the health of individuals who live on Earth [6,7,8,9]. Grounding can reduce pain, regulate immune response, and reduce inflammation-related chemical factors, which are all beneficial for the treatment of inflammatory and autoimmune diseases [10]. Additionally, better sleep quality was noted after grounding [11]. A pilot study showed that 1 h of grounding can significantly induce a pleasant mood and relaxation when compared to a sham-grounding group [7]. Grounding has been reported in numerous previous studies to have neuromodulatory effects in the brain and to regulate dysfunction of the nervous system [6,12,13]. A randomized controlled trial showed that grounding can improve not only pain but also depressed mood, tiredness, general health condition, and quality of life [6].

Agitation, apathy, depression, and anxiety may not respond to acetylcholinesterase inhibitors or memantine in AD. However, antipsychotics, antidepressants, sedative drugs or anxiolytics, and antiepileptic drugs, which are typically prescribed, should be decreased in usage or dosage. Although grounding has many positive effects, there has been no research on the benefits of grounding in patients with AD. This study aims to explore grounding as an alternative method for improving BPSD, including poor sleep quality, anxiety, and depressed mood in patients with AD.

## 2. Materials and Methods

### 2.1. Ethics Approval

The study was approved by the Human Ethics Committee of the Chang Gung Medical Foundation Institutional Review Board (CGMH-IRB No201901136B0). All participants provided informed consent before their enrollment in this study. The study has been registered at ClinicalTrials.gov (accessed on 9 February 2022) (Identifier: NCT05246332).

### 2.2. Subjects

We recruited patients who were diagnosed with AD using the Clinical Dementia Rating scale (CDR), obtaining a score of 1, and undergoing treatment with one of the following drugs: donepezil, rivastigmine, or galantamine [14]. They also met the following inclusion criteria: aged >65 years, able to sit with bilateral bare feet on the grounding mat, able to answer the questionnaire, and able to sign the informed consent adequately. The exclusion criteria for this study were based on the following factors: aged less than 65 years, inability to sit with bilateral bare feet on the grounding mat, or currently taking anti-oxidative health supplements or anti-inflammatory medicine. A total of 22 volunteers were recruited for this study.

### 2.3. Study Design

This was a prospective, randomized, double-blind study. This study was conducted at the Department of Neurology and Chinese Medicine at Kaohsiung Chang Gung Memorial Hospital from August 2019 to July 2020. The participants were recruited from the AD day care community. The participants were randomly assigned to either the grounding group (grounding, grounding mat with grounding cord, *n* = 11) or the sham-grounding group (sham-grounding, grounding mat without grounding cord, *n* = 11) using a randomized block design and research randomizer (https://www.randomizer.org, accessed on 25 July 2021). All participants received grounding for 30 min at a time, five times per week for 12 weeks. We used the electrochemical analyzer CHI 1205b and completed the questionnaire before the study (week 0) and 12 weeks after starting grounding. 

### 2.4. Grounding Methods

Grounding can be achieved by direct contact with the Earth’s surface (e.g., by walking barefoot) or by using conductive systems (e.g., grounding mats) [8]. People can connect to the Earth by sleeping or sitting on the grounding mats, which are equipped with conductive wire connecting to the ground outside [10]. The connection to the Earth allows the free electrons to flow into the body and reach electrical equilibrium between the body and the Earth [11]. In our study, we used a grounding mat (EARTHING Conductive Earthing Universal Mat with Earthing Cord) as our grounding method, which is convenient for studying grounding in the community and prevents the participants from agitation or discomfort in an unfamiliar environment. We used a grounding wire and a sham grounding wire to create grounding and sham-grounding groups. Participants sat on a chair barefoot in contact with the grounding mat for 30 min. The participants were blinded to the grounding since they did not know which one had a grounding effect according to the exterior of the mat.

### 2.5. Blinding Procedures

Each participant’s group assignment was placed in a nonpermeable bag, and both interventions were represented with a number code. The participants, study staff, and investigators were blinded to the group assignment and interventions. The staff who performed the electrochemical analysis and administered the questionnaires were all blinded. The code and group assignment were only known to the person who prepared the grounding mat. After the last participant finished the study, the number code was revealed to the investigator for data assessment. 

### 2.6. Outcome Measures

#### 2.6.1. Method of Electrochemical Analyzer

In a previous study, grounding was shown to permit mobile electrons to enter the body and neutralize free radicals [15]. An electrochemical analyzer is a device based on amperometry that can measure the current in the electrode by a reduction–oxidation reaction [16]. Although the electric signals are small during the redox cycle, they can be amplified during the electrochemical–chemical redox cycle and to analyze the reduction–oxidation reaction measured by voltammetry [17]. 

A CHI 1205b electrochemical analyzer (CH Instruments, Inc. Austin, TX, USA) equipped with Ag–AgCl electrodes (VIVO METRIC, Healdsburg, CA, USA) was used in this study. The electric current unit interconversion between the moles of electrons and moles of superoxide is defined as one ampere equals 10.36 × 10^−6^ moles of electrons per second. One mole of superoxide produces one mole of electrons; thus, one ampere is equal to 10.36 × 10^−6^ moles of superoxide per second. The electrodes were placed on the skin of kidney meridians 1 (KI1) and governor vessel 16 (GV16) according to the World Health Organization standardized acupuncture point locations [18].

#### 2.6.2. Cognitive, Sleep, Anxiety, and Depression Evaluation

The Mini-Mental State Examination (MMSE) was used to evaluate the general intellectual functioning [19]. Cognitive severity was defined using the CDR [20]. The Pittsburgh Sleep Quality Index (PSQI) has 19 self-rated questions which are combined to form seven components; it was used as a screening tool for sleep quality evaluation in this study [21]. Each question for sleep quality was divided into 0 to 3 points according to severity. We summed the PSQI scores to compare the sleep quality. The Chinese versions of the Beck Anxiety Inventory (BAI) and Beck Depression Inventory-II (BDI-II) are valid tools for evaluating anxiety and depression, respectively [22,23]. The BAI has 21 items involving 4 components of anxiety (i.e., autonomic, neurophysiological, panic, and subjective) and 0 to 3 points according to the degree of anxiety in each item [22]. The BDI-II has 21 items with 0 to 3 points for each item, and a higher score is positively correlated with severity [23].

### 2.7. Data Analysis

We retrospectively calculated the power of the sample size. G Power software was used to calculate the power, with the result being equal to 0.93 when the mean ± standard deviation was 3.0 ± 1.9 in the control group and 0.3 ± 0.7 in the grounding group, as compared by independent *t* test under a two-sided significance level of 0.05. 

The age, MMSE, PSQI, BAI, BDI-II, and value detected by the CHI 1205b electrochemical analyzer are presented as the mean ± standard deviation and were compared using the Mann–Whitney U test. We compared sex and comorbidities, including diabetes mellitus, hypertension, hyperlipidemia, and coronary artery disease, using Fisher’s exact test to analyze the differences between groups.

We used a generalized estimating equation (GEE) to check the relationship between group (sham-grounding and grounding), acupoint position electrochemical signals (KI1 and GV16), and timing (week 0 and week 12). To understand the effect of grounding, the acupoint position electrochemical signals and timing of grounding with sleep quality, anxiety, and depression were utilized. Furthermore, the GEE was used to estimate the parameters of the above factors and allowed the PSQI, BAI, and BDI-II to be analyzed separately as outcomes.

Statistical significance was considered at a *p*-Value of <0.05. All statistical analyses were performed using SPSS for Windows, Version 19 (Statistics 19, SPSS, IBM Corp., Chicago, IL, USA).

## 3. Results

### 3.1. Demographic and Baseline Data of Participants 

Fifteen subjects completed the 12-week study course (seven subjects in the sham-grounding group and eight in the grounding group). The mean ages of the enrolled subjects were 77.3 and 80.0 years old in the sham-grounding group and grounding group, respectively. There were 4 men and 11 women. Age, sex, and MMSE scores did not differ significantly between the sham-grounding and grounding groups. The comorbidities included type 2 diabetes mellitus, hypertension, hyperlipidemia, and coronary artery disease, which were not significantly different between the two groups (Table 1). 

### 3.2. Superoxide Concentrations at KI1 and GV 16 

The mean and standard deviation values of the superoxide concentration at KI1 before grounding were 50.18 ± 58.13 and 30.11 ± 10.85 in the sham-grounding group and the grounding group, respectively, without statistically significant difference (*p* = 0.755). After grounding for 12 weeks, the concentration in the sham-grounding group was −76.17 ± 383.03, and that in the grounding group was −173.96 ± 522.06, without statistically significant difference (*p* = 0.755).

At the GV16 acupoint, the mean and standard deviation values of the superoxide concentration before grounding were 27.45 ± 85.98 and 15.80 ± 32.67 in the sham-grounding and grounding groups, respectively, without a statistically significant difference (*p* = 0.228). After grounding for 12 weeks, the concentrations were −37.17 ± 230.56 in the sham-grounding group and −113.58 ± 294.76 in the grounding group, yet the difference was not statistically significant (*p* = 0.414) (Table 2).

### 3.3. GEE for the Effects of Grounding, Acupoint Electrochemical Signals, and Timing 

In the group comparison, there was no significant difference between the groups (*p* = 0.509). In the acupoint comparison, there was no statistically significant difference in terms of position (*p* = 0.760). For the different timepoints, the *p*-value was 0.066 after 12 weeks of grounding (Table 3). 

### 3.4. Sleep Quality, Anxiety, and Depression among Subjects before and after Grounding 

There were no statistically significant differences in the PSQI, BAI, and BDI-II scores between the groups at baseline. After grounding for 12 weeks, the PSQI score was significantly lower in the grounding group than in the sham-grounding group (*p* = 0.006). Furthermore, the subjects with better sleep quality presented a decreased PSQI score in the grounding group than in the sham-grounding group (62.5% vs. 14.3%, *p* = 0.031). The BAI and BDI-II scores did not show a significant difference between the sham-grounding and grounding groups after 12 weeks of grounding (*p* = 0.613, *p* = 0.189) (Table 4).

### 3.5. GEE for the Effects of Grounding, Acupoint Electrochemical Signals, and Timing in Terms of the PSQI, BAI, and BDI-II 

We used the GEE to estimate the parameters of the group (sham-grounding and grounding), acupoint position electrochemical signals (KI1 and GV16), and timing (week 0 and week 12) regarding the PSQI. The *p*-value of grounding was 0.053 in estimating the PSQI (Table 5, Table 6 and Table 7). 

We further used the GEE to estimate the parameters of the group (sham-grounding and grounding), acupoint position electrochemical signals (KI1 and GV16), and timing (week 0 and week 12) regarding the BAI. The *p*-value of grounding was 0.260 in estimating the BAI.

Finally, we used the GEE to estimate the parameters of the group (sham-grounding and grounding), acupoint position electrochemical signals (KI1 and GV16), and timing (week 0 and week 12) regarding the BDI-II. The *p*-value of grounding was 0.394 in estimating the BDI-II.

## 4. Discussion

The main finding of the present study is that grounding can improve average sleep quality according to diminished PSQI scores. However, grounding had no positive effect on anxiety and depression in patients with dementia. The acupoint electrochemical signals at KI1 and GV16 showed no obvious difference before and after completion of the 12-week study. 

In this study, the electrochemical analysis showed a positive charge before grounding in both groups. This result is similar to that of a previous study in which it was implied that losing contact with the Earth may contribute to electric unevenness, causing positive charges and electron deficiency in the body [15]. Grounding can affect physiological human processes and increase catabolic activity, which may be the primary factor regulating endocrine and nervous systems [24]. Superoxide accumulates in the human body and results in dysfunction or disease when the body part disconnects from the Earth [15]. It is hypothesized that electrons flow into the human body from the Earth to neutralize positively charged free radicals after grounding, which is the hallmark of chronic inflammation [25,26,27]. Based on the results of previous studies [25,26,27], we used an electrochemical analyzer to detect the superoxide concentration and electric current at the acupoints. However, electric current flow toward the Earth was observed in both groups and at both acupoints after the 12-week experiment. There was no significant difference between the grounding and control groups, although the tendency of the superoxide concentration seems larger with the current flowing towards the Earth in the grounding groups compared to the sham-grounding group (Table 2). We used a GEE to correct the possible standard error and to investigate the relationship between both groups, acupoints (GV16 and KI1), and timepoints (baseline and 12 weeks). There was no significant difference between the dependent variables in our study (Table 3). However, the *p*-value for time was 0.066 (Table 3). We believe that if more subjects were enrolled, there might have been a significant result after 12 weeks of grounding. 

Our results could not fully explain that electrons flow into the body to neutralize positively charged free radicals after grounding as in previous studies [25,26,27]. We believe that the behavior and daily activity of humans are different, unlike those in animal experiments. Grounding is a simple method that involves direct contact of the body with the Earth, including sleeping on the ground, barefoot gardening, or direct contact with the soil via the hands [15]. In our study, we could not monitor the patients outside of the experiment during the 12-week study. The inconsistency may therefore be due to accidental grounding in other ways, such as falling down with the hands or the body contacting the ground. 

Sleep quality was significantly improved in the grounding group in our study. In non-pharmacologic therapy for AD, light therapy was reported to improve PSQI score, total sleep time, and sleep efficiency by regulating the circadian rhythms of cortisol levels [28,29]. A previous study showed that grounding the body can improve sleep quality by normalizing diurnal cortisol rhythms [30]. In a blinded pilot study, most grounded subjects experienced symptomatic improvement in feeling rested upon waking, sleep quality, and time taken to fall asleep; the results revealed a realignment and normal trend of circadian cortisol patterns after 6 weeks of grounding [11]. Our results are similar to those of a previous grounding study; it is believed that the mechanism of grounding improves circadian rhythms of cortisol due to functioning as a form of light therapy.

Anxiety and depression did not improve after grounding. Anxiety and depression are some types of BPSD [31]. The mechanism of BPSD is associated with serotonergic system dysfunction in the hippocampus and prefrontal cortex [32]. Both glucocorticoids and serotonin can regulate circadian rhythms of cortisol via the suprachiasmatic nucleus [33,34], and the dysregulation of the circadian rhythm is often associated with several mental disorders, such as anxiety and depression [35]. Although no study has applied grounding to the BPSD of dementia, a previous study showed that grounding reduces stress and improves sleep quality, which can further improve anxiety and depression in clinical observation [15]. Grounding has been proven to improve cortisol-related sleep quality, anxiety, depression, and irritability [30]. Grounding for 1 h can improve pleasant and positive moods in healthy participants compared to those in a control group [7]. It was also found that participants felt less emotional stress, such as anxiety, depression, or irritability, after grounding, which may be related to the normalization of cortisol and circadian levels in the body [30]. Hence, we hypothesized that grounding has the potential to improve anxiety and depression by improving circadian rhythm. Nonetheless, the sample size of this study was too small to show an effect of grounding on anxiety and depression.

The major strength of this investigation is that it is the first study to investigate the effect of grounding in patients with diagnosed dementia, specifically in the AD day care community. Benefits of grounding were found; however, there are some limitations to this study. The first limitation of our study was the small sample size, which may have resulted in bias and non-significant differences between the groups. This indicates that a large sample size is needed to demonstrate the effect of grounding. Second, grounding is a simple method that can be easily achieved. Therefore, we suggest designing different periods of time, such as 8 weeks, 12 weeks, 16 weeks, and 20 weeks, to compare the effects of different exposure times in the future. Third, we tried our best to use only a grounding mat for easy grounding and randomizing methods, and we asked the patients to not walk outside barefoot. Nonetheless, the participants might have accidentally participated in grounding in their daily life when their feet were in contact with the Earth, such as soil or grass.

## 5. Conclusions

Grounding is beneficial for improving sleep quality in patients with dementia. However, depression and anxiety did not improve after grounding, and it is possible that the duration of the grounding might be correlated with its effect. Further studies are needed to explore the effect of grounding over longer periods of time, and a larger sample size of participants is called for.

## Figures and Tables

**Table 1 healthcare-10-00581-t001:** Demographic and baseline data of participants *(n* = 15).

	Control Group (*n* = 7)	Grounding Group (*n* = 8)	*p*
Age	77.3 ± 9.6	80.0 ± 6.9	0.414
MMSE	17.6 ± 6.8	16.2 ± 4.4	0.449
CDR	1.0	1.0	-
Gender (m/f)	2/5	2/6	0.662
Type2 DM, n (%)	4 (57.1%)	3 (37.5%)	0.231
Hypertension, n (%)	4 (57.1%)	4 (50%)	0.595
Hyperlipidemia, n (%)	1 (14.3%)	1 (12.5%)	0.733
CAD, n (%)	1 (14.3%)	1 (12.5%)	0.733

Values are expressed as the mean ± standard deviation. A Mann–Whitney U test was used to analyze the difference between groups; a Fisher’s exact test was used for analysis of the categorical variables. Abbreviations: MMSE, Mini-Mental State Examination; CDR, Clinical Dementia Rating; DM, Diabetes Mellitus; CAD, coronary artery disease.

**Table 2 healthcare-10-00581-t002:** Comparison of superoxide concentrations at kidney meridians KI 1 and governing vessel GV 16.

	Control Group (*n* = 7)	Grounding Group (*n* = 8)	*p*
Kidney Meridians KI 1 electrochemical signals			
Baseline	50.18 ± 58.13	30.11 ± 10.85	0.755
12 weeks	−76.17 ± 383.03	−173.96 ± 522.06	0.755
Governing Vessel GV 16 electrochemical signals			
Baseline	27.45 ± 85.98	15.80 ± 32.67	0.228
12 weeks	−37.17 ± 230.56	−113.58 ± 294.76	0.414

Values are expressed as the mean ± standard deviation. The Mann–Whitney U test was used to analyze the differences between groups (×10^−15^ mole).

**Table 3 healthcare-10-00581-t003:** Generalized estimation equation for kidney meridians KI 1 and governing vessel GV 16 superoxide concentrations after the indicated grounding period (study and control groups) (×10^−15^ mole).

			95% Wald Confidence Interval		Hypothesis Test		
	B	Standard Error	Lower	Upper	Wald Chi Square	df	Sig.
Group							
Control	0						
Grounding	−51.477	78.0101	−204.374	101.420	0.435	1	0.509
Acupoint electrochemical signals							
Kidney Meridians KI 1	0						
Governing Vessel GV 16	16.654	54.6064	−90.373	123.680	0.093	1	0.760
Time							
Baseline	0						
12 weeks	−136.189	73.9816	−281.1191	8.812	3.389	1	0.066

**Table 4 healthcare-10-00581-t004:** Comparison of sleep quality, anxiety, and depression among subjects before and after grounding.

	Control Group (*n* = 7)	Grounding Group (*n* = 8)	*p*
Baseline			
PSQI	2.0 ± 2.1	1.9 ± 1.9	1.000
BAI	1.7 ± 1.4	0.8 ± 0.9	0.228
BDI	7.3 ± 3.9	5.1 ± 3.5	0.081
12 weeks			
PSQI	3.0 ± 1.9	0.3 ± 0.7	0.006 *
BAI	0.9 ± 1.4	0.6 ± 1.4	0.613
BDI	7.2 ± 4.1	5.3 ± 7.0	0.189
Score decrease after 12 weeks of grounding			
PSQI, *n* (%)	1 (14.3%)	5 (62.5%)	0.031 *
BAI, *n* (%)	3 (42.9%)	4 (50%)	0.782
BDI, *n* (%)	1 (14.3%)	4 (50%)	0.1432

Mean ± Standard deviation (Mann–Whitney U test and Fisher’s exact test). * *p* < 0.05, vs. control group. Abbreviations: PSQI, Pittsburgh Sleep Quality Index; BAI, Beck Anxiety Inventory; BDI, Beck Depression Inventory.

**Table 5 healthcare-10-00581-t005:** Generalized estimation equation for the PSQI.

			95% Wald Confidence Interval		Hypothesis Test		
	B	Standard Error	Lower	Upper	Wald Chi Square	df	Sig.
Group							
Control	0						
Grounding	−1.407	0.7278	−2.834	0.019	3.738	1	0.053
Acupoint electrochemical signals							
Kidney Meridians KI 1	0						
Governing Vessel GV 16	−0.010	0.0346	−0.078	0.058	0.081	1	0.776
Time							
Baseline	0						
12 weeks	−0.0419	0.6444	−1.682	0.843	0.424	1	0.515

**Table 6 healthcare-10-00581-t006:** Generalized estimation equation for the BAI.

			95% Wald Confidence Interval		Hypothesis Test		
	B	Standard Error	Lower	Upper	Wald Chi Square	df	Sig.
Group							
Control	0						
Grounding	−0.561	0.4977	−1.536	0.414	1.271	1	0.260
Acupoint electrochemical signals							
Kidney Meridians KI 1	0						
Governing Vessel GV 16	0.000	0.0079	−0.016	0.015	0.004	1	0.952
Time							
Baseline	0						
12 weeks	−0.425	0.4604	−1.327	0.478	0.851	1	0.356

**Table 7 healthcare-10-00581-t007:** Generalized estimation equation for the BDI.

			95% Wald Confidence Interval		Hypothesis Test		
	B	Standard Error	Lower	Upper	Wald Chi Square	df	Sig.
Group							
Control	0						
Grounding	−2.167	2.5421	−7.150	2.815	0.727	1	0.394
Acupoint electrochemical signals							
Kidney Meridians KI 1	0						
Governing Vessel GV 16	0.034	0.1222	−0.206	0.273	0.077	1	0.781
Time							
Baseline	0						
12 weeks	−0.277	0.5209	−1.298	0.744	0.283	1	0.595

## Data Availability

Not applicable.

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
