# Peer review of "Grounding the Body Improves Sleep Quality in Patients with Mild Alzheimer’s Disease: A Pilot Study"

_healthcare, 2022, doi:10.3390/healthcare10030581_

Round 1

Reviewer 1 Report

This paper entitles "Grounding (Earthing) the body improves sleep quality in patients with mild Alzheimer's disease" aimed to investigate the effects of grounding in patients with mild Alzheimer's disease and concluded that the grounding is effective in improving sleep quality. This is an interesting manuscript that gives insights on how the Grounding may be used for treating diseases.  There are however issues that must be addressed to improve the clarity and manuscript readability.

1- Title, considering the study limitations, I would suggest the study be considered as a pilot study, thus the title should be revised accordingly. I would suggest omitting the "(Earthing)" from the Title.

2- The introduction is well-written justifying why the study was required to perform. 

3- Methods, page 2, Study design should be clearly reported. Please report the registration number for this study.

4- Page 3, line 101 is repetition with page 2, lines 89-90.

5- Page 3, before "Assessments", I would suggest clearly reporting the dependent variables under the "Outcome measures".

6- Assessments should report on the assessor/s blinding. 

7- Page 3, line 115 should be referenced.

8- Page 3, line 119, please insert "the" before "general".

9- Page 3, line 122, please report the number of items for PSQI.

10- Page 3, Data analysis, line 132, what do you mean by "mean deviation? With regard to the small sample size, I would suggest the analyses should use non-parametric statistical tests. Also, please report the effect size for the outcomes if significant changes were found.

11- Page 3,  lines 133-135 need clarification. It is a concern that the sample size is not prospectively calculated. 

12- Results section needs to be rewritten after performing new analyses. Also, please do your best to well organize and present the results.  Please report the results first for the grounding group followed by the control group in the text as well as in the Tables. 

13- Page 4, Table 2 is a repetition with those of the text, thus could be omitted. Pages 5-7, results and tables should avoid repetitions. Tables, abbreviations should be defined (e.g. Table 4). 

14- Discussion should be rewritten considering new results with non-parametric tests. 

15- Discussion, page 7, line 239, you mean "initial hypothesis"? Please clarify it.

16- Discussion, page 7, line 246, I do not tink falling down accidentally for short duration to be adequate for grounding effects.

17- Discussion, page 8, line 256, I do not think the grounding to be considered as a type of light therapy.

Reviewer 2 Report

Overall, this study is logically well described. But I have some recommendations.

As you mentioned, this study was performed from August 2019 to July 2020. During the study period, it happened the coronavirus outbreak. Was there any direct impact on this study? It seems that it may have had an effect not only difficulties in research performance, but also on depression and anxiety.

In this study, G-power was used to determine the sample size. Please provide the rationale to determine the sample size.

Reviewer 3 Report

  1. The research content of this paper is relatively novel, but how the grounding mat realizes the grounding effect should be explained clearly.
  2. “Grounding refers to having direct contact with the Earth, such as walking barefoot or lying on the ground. ”Is the effect of connecting the “grounding mat” the same as that of actually contacting the ground (such as walking barefoot)? Is there any evidence for this?
  3. This study has a limited impact since the sample size was very small(only 15patients). The second paragraph of the discussion also mentioned “we thought that if there were more subjects enabled, there might have been a significant result after 12 weeks of grounding”

Round 2

Reviewer 1 Report

I appreciate the authors for revising the manuscript. It now looks much better. 

Reviewer 3 Report

The manuscript has been modified according to the comments of reviewers and is recommended to be accepted.